# Selective Light Measurement in the Control of Reference LED Sources

**DOI:** 10.3390/s23063285

**Published:** 2023-03-20

**Authors:** Marian Gilewski

**Affiliations:** Faculty of Electrical Engineering, Bialystok University of Technology, 45A Wiejska Street, 15-351 Bialystok, Poland; m.gilewski@pb.edu.pl

**Keywords:** optical Micro Electro-Mechanical Systems, field-programmable gate array control, electrical signal conditioning

## Abstract

This paper describes an original adaptive multispectral LED light source that utilizes miniature spectrometer to control its flux in real time. Current measurement of the flux spectrum is necessary in high-stability LED sources. In such cases, it is important the spectrometer work effectively with the system that controls the source and the whole system. Therefore, as important as flux stabilization is the integration of the integrating sphere-based design with the electronic module and power subsystem. Since the problem is interdisciplinary, the paper mainly focuses on presenting the solution of the flux measurement circuit. In particular, the proprietary way of operating the MEMS optical sensor as a real-time spectrometer was proposed. Then, the implementation of the sensor handling circuit, which determines the spectral measurements accuracy and thus the output flux quality, is described. Also presented is the custom method of coupling the analog part of the flux measurement path with the analog-to-digital conversion system and the control system based on the FPGA. The description of the conceptual solutions was supported by the results of simulation and laboratory tests at selected points of the measurement path. The presented concept allows to build adaptive LED light sources in the spectral range from 340 nm to 780 nm with adjustable spectrum and flux value, with electrical power up to 100 W, with adjustable flux value in the range of 100 dB, operating in constant current or pulsed mode.

## 1. Introduction

For more than half the century, light-emitting diodes (LEDs) have been an integral part of everyday life. Initially, their importance was insignificant, but technological developments have led to increasing applications in many fields [1,2,3,4,5,6,7]. LED illuminators have many advantages over traditional incandescent or discharge light sources. Such advantages include the ability to synthesize colors, wide emission angles, high contrast and high luminous efficacy, low-voltage power supply and user-friendly flux control techniques [8,9,10]. Unfortunately, LED sources also have specific drawbacks, such as the LEDs binning, the LED aging, ambient temperature, junction temperatures, electronic driving method, operating time and ambient humidity [11,12,13]. However, the high efficiency, design advantages of LEDs and their easy power supply have made them increasingly widely used. With the development of LED technology, at first, we attempted to stabilize the operating point of single LEDs and complete LED sources. The following researches were not limited only to thermal and current stabilization, but also included multispectral synthesis attempts [14]. Choosing the right lighting is fundamental in optical laboratory measurements, including colorimetric measurements. Consequently, scientific research has also been concerned with the development of LED reference sources that would comply with CIE (Commission Internationale de l’Éclairage) requirements [15]. The primary design challenge in multispectral LED sources is integrating monochromatic fluxes into one coherent output. In the accessible knowledge sources, two methods for synthesizing reference sources can be found: the method based on an integrating sphere [16,17] and the method with homogenizing rods [18]. Since the applications of these methods work in open control mode, they cannot compensate the factors that destabilize the LED illuminator. Only the feedback control system can provide fully adaptive illuminator operation in real time.

This article presents the concept of the multispectral LED source with feedback, which can be a lamp hardware simulator. The very important component of such a source is the return measurement system of the flux value and its spectrum. The development of the feedback measurement system was the main objective of this work.

The inherent component of the optical feedback measurement system in each multispectral source is the spectrometer type. Therefore, the following section describes the spectrometer development and evaluates its applicability in the designed system. Integration of components and sensor miniaturization and development of distributed measurement systems are integral features of modern measurement systems. These trends also apply to optical radiation metrology, and open the way for new applications [19]. Traditional spectrometers (Figure 1a) typically include a one- or two-stage monochromator, a photo detector and a computer control system. The stationary nature of these instruments does not enforce either size optimization or energy savings. Therefore, there are no design limitations in stationary apparatus, making it possible to achieve high spectral resolution and build instruments for a wide range of measured flux values. These desktop systems are and will continue to be high-quality reference measurement instruments. Their disadvantage is low mobility. Therefore, they are completely unusable in cases where the spectrometer is part of the measurement system, and it measures the online return signal. In such using, not only the dimensions are a barrier, but also the time inertia. Some LED sources produce pulsed flux as high as 10 kHz, so scanning time of the stationary spectrometer of approximately 1 s is unacceptable.

In chosen applications, the successors of stationary spectrometers have become mini-spectrometers (Figure 1b). They used miniature monochromators that were integrated with CCD arrays. Such technology allows the construction of more mobile measurement systems. However, such solutions are characterized by lower measurement resolution, and their measurement results are affected by higher values of resulting errors [20]. Sources of measurement errors in mini-spectrometers include dark current, thermal sensitivity, stray light, photoelectric conversion nonlinearity and electronic circuit errors [21]. In some designs of mini-spectrometers, stray light attenuation is approximately−30 dB or −25 dB, which classifies them in the middle class of spectrometers. The undoubted advantage of these spectrometers is the shorter measurement time and the elimination of mechanically driven monochromators. It results in low power consumption and increases the instrument’s resistance to mechanical vibration. However, in such construction, all the photoelectric signals flow within the enclosed housing of the monochromator, and the user cannot use these signals to control the LED illuminator. Therefore, this technology is of limited utility in control systems, too. The controllability limitations using spectrometers have been eliminated by the latest integrated solutions MEMS (Micro Electro-Mechanical Systems) optical sensors [5,6,7,22,23,24]. In this sensor category, all spectrometer components are integrated into a single miniature hybrid chip (Figure 1c). This results in the miniaturized chip that measures flux over the entire visible range in user-controlled mode. It provides the Video analog output signal, the amplitude of which is proportional to the flux values in 12nm intervals distributed between 340 nm and 780 nm wavelengths. The MEMS spectrometer is powered by a low 5 V DC, and it requires several digital control signals to operate. The sensor module is enclosed in the shielded and sealed metal case with no moving parts [25]. This makes the module resistant to vibration and electromagnetic interference. This metallic case allows temperature stabilization of all MEMS components, which is not achievable in desktop spectrometers and mini-spectrometers.

As can be seen in Figure 2, the MEMS sensor synchronously outputs for successive spectral intervals of 12 nm the measured flux value in the form of Video pulses.

**Figure 1 sensors-23-03285-f001:**
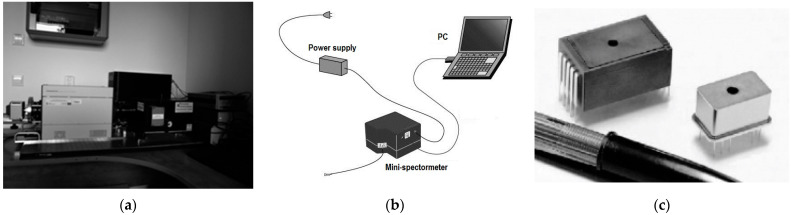
Spectrometer architectures have evolved: (**a**) a traditional spectrometric system; (**b**) mini-spectrometer architecture; (**c**) MEMS spectrometer [25].

Thus, the amplitude of the first Video pulse has a coded flux value, for example, in the band from 340 nm to 352 nm. The amplitude of the particular pulse also depends on the integration time, which can be controlled by the clock signal frequency. This technology requires the Clock frequency to be between 1 kHz and 800 kHz. A single Video pulse is transmitted in four consequential clock pulses. The measurement cycle is initiated by the master device, which is a microcontroller or FPGA. It sends the Start signal, while the spectrometer ends the measurement cycle by setting the End of scan signal. According to the dynamics shown in Figure 3, the spectrometer can measure the input flux over a very wide range, from 0.5 pW to approximately 0.5 μW.

With such large changes in the input signal, adjustable signal gain in the measurement path is required, which should be as high as 120 dB. Measurement circuit architecture is additionally complicated by the signal Video nature, which is a series of rectangular pulses tunable in frequency. Therefore, this paper proposes its own modified circuit to conditioning the output Video signal before that signal is converted to digital representation. Of the spectrometer concepts discussed earlier, the MEMS spectrometer is the most suitable for use in feedback of the LED source circuit, particularly the reference source.

## 2. Materials and Methods

The motivation for the present work was the unsatisfactory state of knowledge in the field of applied research on semiconductor light standards and LED special lamps. Although basic research in this field is highly developed, the state of applied research is sometimes unsatisfactory. Since application research is followed by development work, it often results in products that do not meet assumptions and expectations [7].

This unfavorable condition is due to the product design process. Very often, the research and design cycle typically includes the following stages:Selection of available and low-cost LEDs;Development of the control system for the LEDs;Prototype design;Running the completed prototype;Production of a test series;Completing the test installation;Final testing of lighting effectiveness.

In such a product development cycle, there is no space for hardware verification of illuminator parameters, including spectral matching at the design stage. Therefore, there is the need for developing the hardware simulator of the luminance model which could act as a tunable standard to verify the results of LED illuminator design as they occur.

Such a reference source model should be stable and tunable, both in the wavelength space and luminous flux values. These properties are provided only by an LED array illuminator, which contains a wide collection of monochromatic LEDs operating in adaptive mode. This means that the system should have the feature of automatically correcting the operating points for all LEDs included in the illuminator. Therefore, the LEDs’ corrected currents must be related to the spectral measurement of current source flux. The conditions of real-time flux measuring and the using the measurement results at the electrical signal level are provided only by a MEMS spectrometer. In this matter, mini-spectrometer architectures or stationary spectrometers are useless. There are few scientific publications on the sensor applications in the feedback path of a controlled system, and even fewer studies describing MEMS spectrometer applications in feedback loops of a lighting system or light source. Moreover, the available publications describe the spectrometer application in general and do not analyze in detail its performance in the measurement circuit. Sometimes, the described application of a MEMS spectrometer includes sensor drivers whose parameters are irrelevant to the described application, and control timing characteristics are omitted [26] or inaccurate. In this paper, there is proposed own sensor handling system, which can handle MEMS spectrometer in such applications more optimally.

The proposed sensor handling system is the component in the complex system (Figure 4), which is the tunable LED reference source. Besides it, the source architecture includes thermally stabilized LED array, FPGA module and mains power supply. The remaining instruments, which are reference spectroradiometer, multichannel controller and PC, do not belong to the source system. They are not required during routine source performance but are used in the preliminary phase, during source calibration. The meaning of the colored connections in Figure 4 is the following: red lines show component supply, yellow connections are signal and data buses, flux flow is indicated by purple signals and the blue network refers to signal flow in the calibration phase.

In the presented solution, the LED array included 25 diodes whose spectra uniformly covered the wavelength range from 275 nm to 780 nm. These diodes were low-power, with supply currents no higher than 100 mA. The LED matrix was mounted on the aluminum disc (Figure 5), which was bonded with the larger base plate.

The base plate additionally included the temperature sensor and the Peltier module for interfacing with the active cooling module. The tested array did not require external cooling, but for higher-power illuminators, an active cooling system is advisable, especially when single diode or diode branch powers exceed several watts. The base plate was directly attached to the input of the integration sphere, which at the same time acted as a chassis for the entire system (Figure 6). On the right side of the matrix base plate, the MEMS spectrometer board can be seen. It contains, besides the optical sensor, the necessary auxiliary components such as the power supply circuit, the sensor output signal conditioning circuit and the analog-to-digital conversion circuit. The MEMS sensor is installed so that its optical input faces the integrating sphere’s center.

The source system architecture shown in Figure 6 is not the final form, but a development version at the physical model level. Therefore, the simplified module of tunable current sources that power the LED array was used. It is an array of the replaceable series resistors, which is shown in this figure’s lower left corner. Finally, the multi-section source set was used; the single section diagram is shown in Figure 7. The tunable current source shown in this figure is based on the current mirror design (e.g., [26]), in which the current Iled is several hundred times the amplified reference current Iref.

According to Ohm’s law, Iref current is a quotient of Uref by Rref, where Uref is the internal reference voltage and Rref is the digital potentiometer resistance. Thus, the FPGA, by setting Rref digitally, can control the current value flowing through a single LED or a LED branch connected in series. In the tested model, the target power supply was replaced with the laboratory power supply, and the FPGA module was substituted with the prototype laboratory module [27]. The prototype FPGA laboratory module was used only during testing, and it is redundant to the requirements. Therefore, only the FPGA, flash memory and clock generator were used in the final LED source solution.

The FPGA chip used performs: MEMS spectrometer handling, digital thermometer reading, powering the cooler fans and Peltier module and, in addition, performs the real-time data processing algorithm. In the described case, the usage of an FPGA is more advantageous than the application of a microcontroller, because there are many processes in the circuit that need to be carried out in parallel. In the conducted research, only part of the control algorithm was tested, which related to MEMS spectrometer handling. For this purpose, the necessary FPGA software module was developed using the Intel Quartus II Prime Light Edition design platform [28]. The concept of the module was based on a hierarchical structure defined in the VHDL hardware description language using the Finite State Machine technique.

Finally, there are two phases during the running of the system shown in Figure 4: the calibration phase and the real-time adaptive working phase. In the configuration phase, before using the source for the first time, the user sets the expected spectral flux characteristics. For this purpose, using PC software, the multichannel controller settings are changed, and in this way, the current values of each LED branch are set. Each LED branch produces a partial flux in the different spectral range. The partial fluxes of all LEDs add up in an integrating sphere and produce a homogeneous reference beam of light on the sphere’s output. This beam is measured using the stationary reference radiometer. If the spectral and power parameters of the reference beam match the assumptions, the MEMS spectrometer performs its own measurement and the FPGA module saves the measured values in flash memory. These values are the reference settings which the system targets during the adaptive operation phase. The configuration phase can be repeated at any time, after which the system is ready for exploitation.

In adaptive real-time running, all LED source functions are controlled by the FPGA module. At the hardware level, this module sets the currents driving the LED branches, handles the MEMS spectrometer, reads the LED array temperature value, controls the currents feeding the Peltier module and the cooler fan. At the same time, the FPGA carries out the control algorithm, which processes the data from the MEMS sensor, compares the current flux distribution in the integrating sphere with the reference value stored in flash memory, keeps the difference between the reference and measured values within an acceptable interval by changing the LED currents, and fixes the values of the currents supplying the Peltier module and fan. Due to the abundance and multi-threading of the component control processes, only the distributed FPGA structure provides a truly concurrent real-time control implementation.

## 3. Results and Discussion

In the feedback loop in the investigated system, the MEMS spectrometer type C12666MA [25] was used to stabilize the optical working point in the reference LED source. No similar examples of using this micro-spectrometer to stabilize the flux of an LED source built on an integrated sphere were found in the available bibliography. On the other hand, one article [29] was found that describes similar applications of the C12666MA chip but in a different field. The implementation described deals with using the C12666MA chip in controlling the housing room lighting. It is not clear in the part concerning the handling method of this sensor. Therefore, the interested reader is not able to repeat the experiment. Such state was an additional motivation to develop my own sensor control system and write this article.

In the mentioned work, the authors described the adaptive living room lighting system (Figure 8), which included a wireless light measurement module in the feedback. This module consisted of the C12666MA chip, the wideband VIS TEMT6000 sensor with V_λ_ characteristics, a microcontroller and the ZigBee radio transmitter. The measured level and spectral composition of light in the living room was sent to the central ATmega328 microcontroller, which controlled the 8-spectral LED lamp. The measured light was the sum of the radiation from the LED lamp and external sunlight. The authors, controlling the 8 LEDs, attempted to achieve the target spectra: warm and cool white LED light. They adjusted the LED optical powers by changing the times of 700 mA pulses using 8-bit 1 kHz PWM modulation. Measuring the entire spectrum and its wireless transmission required a relatively long time of 300 ms. With such a long cycle, it is difficult to synchronize the measurement module with the light pulses of individual LEDs, the shortest of which with 8-bit 1 kHz PWM modulation was only 4 μs. In addition, the cited work does not specify the solar radiation range in which the described system works stably. The tested space was the living room with large windows, where sunlight significantly interferes with the stabilization of the designated spectrum. This is due to the fact that the external (sunlight) illuminance varies over the wide range: from 0.2 lx on a bright moonlit night to approximately100,000 lx on a sunny day at the equator. The LED lamp under study could produce the radiance of less than 10 watts, while the sun can provide a surface flux density of up to 1000 W/m2 [30]. Therefore, with strong outdoor lighting, sunlight will dominate the LED lamp light - so adjustment the lighting spectrum will be very limited. Because excess radiation cannot be subtracted, it is not possible to shape the light spectrum in the studied living room according to the assumed characteristics of warm and cool white LED light.

Figure 9 shows the measured waveforms at the C12666MA spectrometer outputs, which was mounted on the integrating sphere in the system shown in Figure 6. The yellow waveform shows the Video signal (Figure 2) at the spectrometer output. As can be seen, the MEMS sensor sends the measured flux value in the VIS band from 340 nm to 780 nm in the form of a string of rectangular pulses that repeats periodically. The amplitude of each impulse is proportional to the flux value in the measurement interval (spectral resolution C12666MA [25]) of approximately12 nm. Thus, consecutive pulses in the sequence represent flux values in adjacent measurement intervals. Therefore, the envelope of the Video signal in one period reflects the source spectrum in the VIS band. In the measured case, this was the spectrum produced by only two lighting LEDs in the LED array.

As can be seen in the figure above, the Video signal contains a high DC ingredient, which is not associated with the measured optical signal but with the C12666MA technology, and its level depends on dark current, output offset voltage, and so on. This is a constant level that interferes with the measurement and should be separated from the variable component (Figure 10), especially in measuring weak optical signals. Therefore, the Video output in the C12666MA should not be connected directly to the A/D converter input, but through a filter/amplifier. The C12666MA spectrometer can measure optical signals in the broad range of values, that is, about six decades (Figure 3). Therefore, the Video signal should be processed with a high-speed A/D converter and an adjustable amplifier should be used in the measurement path.

In a system containing a MEMS spectrometer chip, it is important that the integration phase beginning be synchronous with the measured optical signal. Similarly, it is necessary to synchronize the start of the analog-to-digital conversion with the spectrometer’s video pulses. It is preferable if the analog-to-digital processing starts in the middle of the Video pulse duration (Figure 11), which requires synchronization with several μs accuracy.

Taking into account the parameters and handling requirements of the C12666MA spectrometer and the implications from the example described in the literature [29], this paper proposes its own operating configuration for this sensor. In the proposed system, the interfering DC component and the measuring AC pulses were extracted from the output Video signal in the spectrometer. For this purpose, two filters were used: the low-pass filter (Figure 12a) which extracts the DC component, and the high-pass filter (Figure 12b), which filters the AC pulses proportional to the measured interval fluxes. The DC component can be decoupled using the passive first-order low-pass filter. A filter with the −3 dB cutoff frequency is sufficient; the simulation results of such a filter are shown in Figure 13. As can be seen, the slope of this characteristic in the stop band sufficiently suppresses interference coming from the 50 Hz power grid. Since the DC component has a relatively large value, therefore it does not require additional amplification. Thus, after filtering, the DC component can be directly converted by one of the channels of the multi-input AD converter.

More careful signal conditioning is required for the AC component. In the case of the AC component, whose amplitude is proportional to the value of the measured flux, not only filtering but also multi-level amplification is necessary. This situation is complicated by the fact that the amplified signal is rectangular pulses. Therefore, the lower cut-off frequency of the high-pass filter should be less than the lowest frequency of the Video signal, which for the C12666MA chip is 250 Hz. In the studied case, the passive second-order high-pass filter was used, the lower cut-off frequency of which was a few Hz. The high-pass filter output was directly connected to the ripple-curry amplifier input (Figure 14).

The wide-range AC component should not be amplified by the single-stage amplifier. Such an amplifier, in the case of weak light signals, would cause deformation in the slopes of the amplified Video pulses. Therefore, the proposed circuit uses the multi-stage ripple-curry amplifier [31], the schematic of which is shown in Figure 14. It is made up of identical sections with the gain of 2, 4 or 8, which are daisy-chained. At the same time, every section’s output is connected to the different inputs in the multi-channel ADC. Thus, each AD converter channel can convert Video pulses in different amplitudes. The FPGA, which supports the AD converter bus, reads the results from the channel in which the measured signal has the highest amplitude but does not yet saturate the converter input. This combination of amplifier and AD converter allows the measurement path to have wider frequency bandwidth than the single-stage amplifier. At the same time, the circuit structurally performs the automatic gain control function, because depending on the signal amplitude, the FPGA can select the properly amplified signal. The FPGA then back-scales the measured value, that is, divides it by the multiple the single amplifier section gain and the number of sections in the circuit. In digital signal processing, this means discarding the appropriate number of least significant bits of data.

Figure 15 shows the gain results of 10 kHz rectangular pulses in selected points in the ripple curry amplifier. The undistorted pulse view indicates that in this frequency range, the Video signal will be correctly filtered and amplified. The 10 kHz frequency is in the middle of the frequency range that the Video signal in the C12666MA spectrometer reaches.

At the Video rate range limits, the linear distortion of the amplified pulses was slightly higher. In the case of low pulse frequencies (Figure 16a), the AC signal conditioning circuit introduces the differential effect. Therefore, the high-pass filter should have the lowest possible value of the lower cutoff frequency. On the other hand, in the upper frequency range (Figure 16b), the visible distortion of the pulses was caused by the operating amplifiers’ frequency parameters. In the case studied, the gain bandwidth product amplifier was only 10 MHz, and its slew rate did not exceed 7 V/μs. Therefore, an amplifier with the slew rate of at least 100 V/μs should be used in the C12666MA analog path.

## 4. Conclusions

The application of a MEMS spectrometer in the feedback measurement system of compact, reference LED sources appear to be the best solution at present. However, systems of this type require a carefully designed circuit for conditioning the output signal before this signal is transferred to analog-to-digital conversion. Hence, it is necessary to take into account the time characteristics of the spectrometer chip used, to match the analog filter–amplifier circuit to the range of measured signals, and to use proper control of both the MEMS sensor and the amplifier and AD converter. It is important that the control circuit, an FPGA or microcontroller, be as close as possible to the MEMS spectrometer. This type of spectrometer is predisposed to measuring optical DC and slow-change signals. Although its integration times range from a few ms to approximately 10 s, the measurement of pulsed signals requires careful synchronization between the operation of all modules. This is especially relevant for lamps that contain spectrally different LEDs, powered by pulses. Such a situation requires correlating the work of the measurement path with the work of the individual LED power supplies. MEMS technology in light measurement technology is developmental and creates new areas of application, and makes it possible to achieve some measurement and operating parameters more advantageous than in stationary spectrometers or mini-spectrometers.

## Figures and Tables

**Figure 2 sensors-23-03285-f002:**
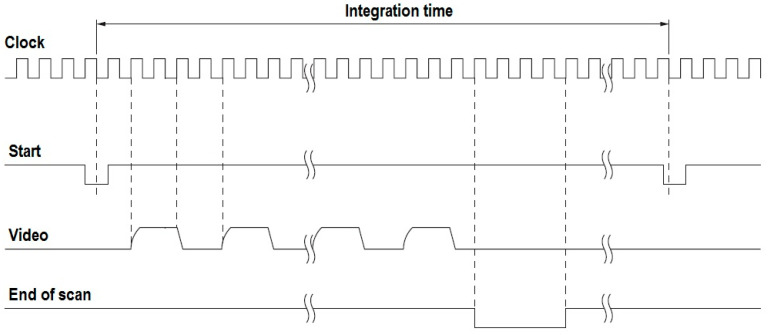
The MEMS theoretical timing charts [25].

**Figure 3 sensors-23-03285-f003:**
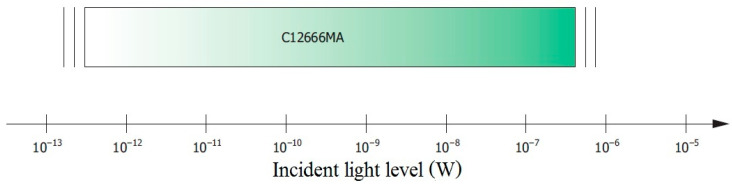
Measurable incident light level on the sensor [25].

**Figure 4 sensors-23-03285-f004:**
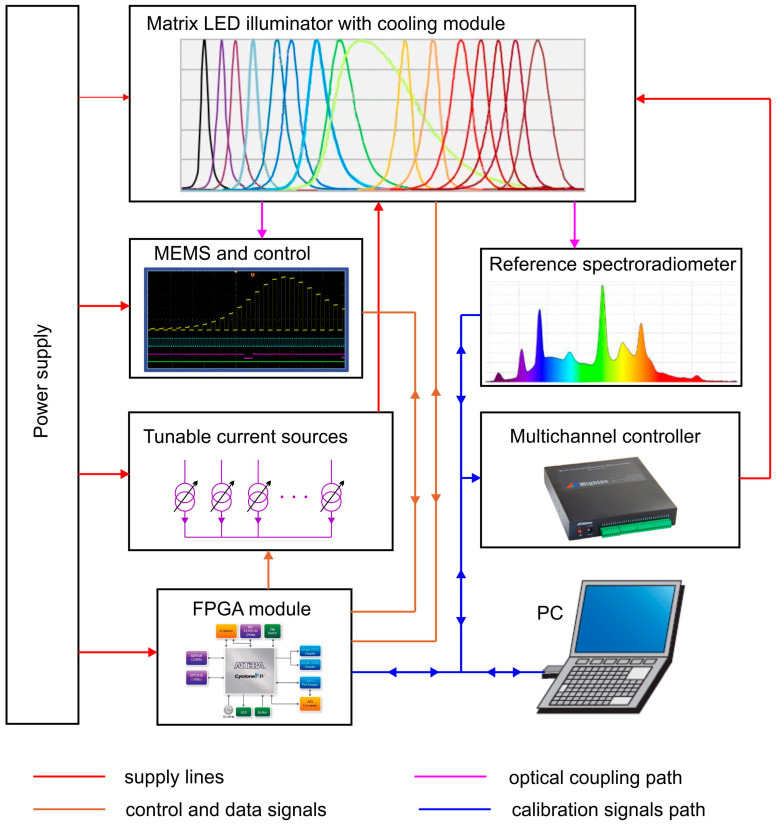
Architecture of the adaptive LED source hardware model.

**Figure 5 sensors-23-03285-f005:**
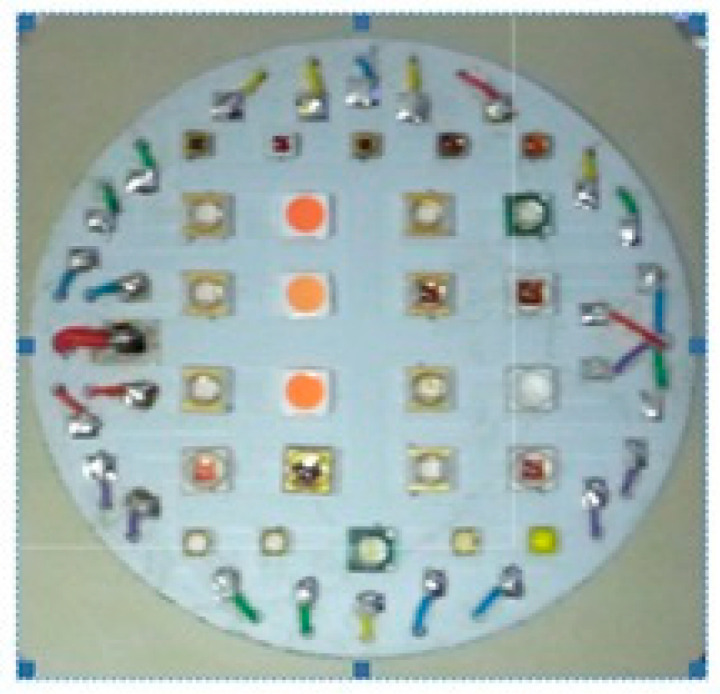
The view of the LED matrix.

**Figure 6 sensors-23-03285-f006:**
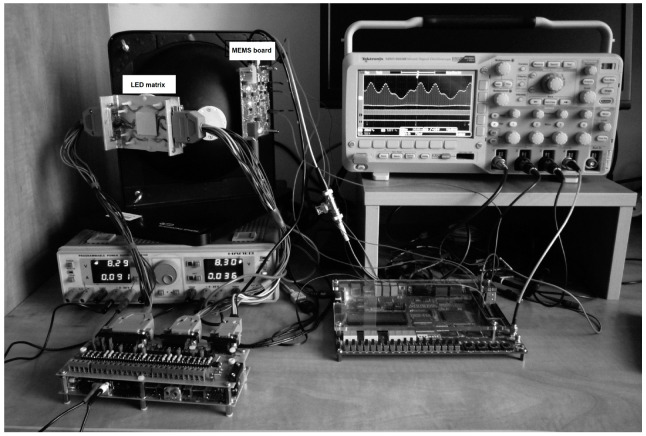
Adaptive LED test platform.

**Figure 7 sensors-23-03285-f007:**
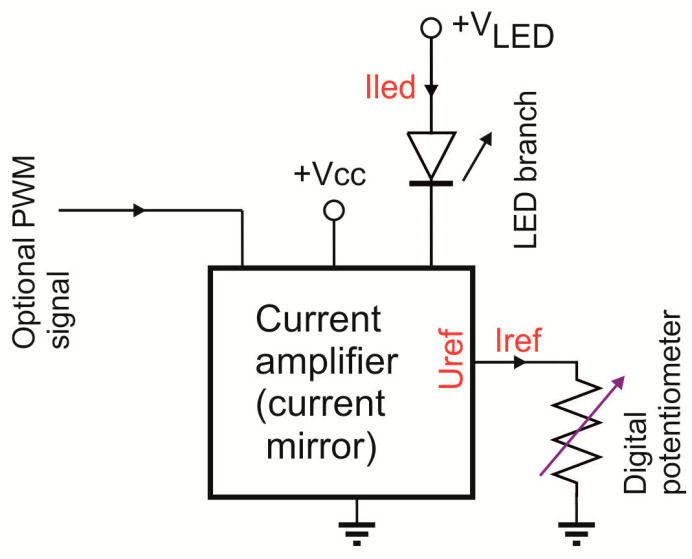
Single LED or LED branch driver.

**Figure 8 sensors-23-03285-f008:**
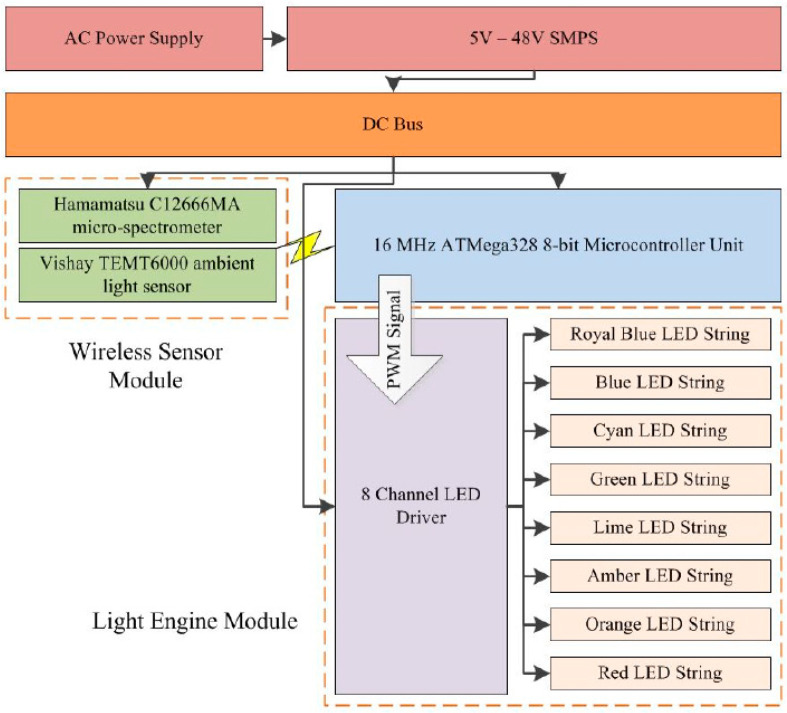
The block diagram of the lighting system [29].

**Figure 9 sensors-23-03285-f009:**
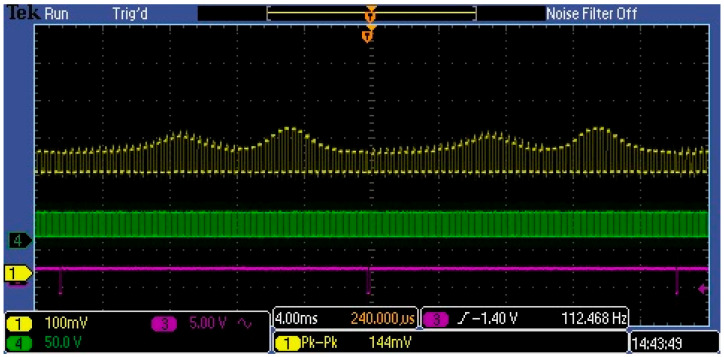
Captured MEMS signals: yellow is Video, green is Clock and purple is End of scan.

**Figure 10 sensors-23-03285-f010:**
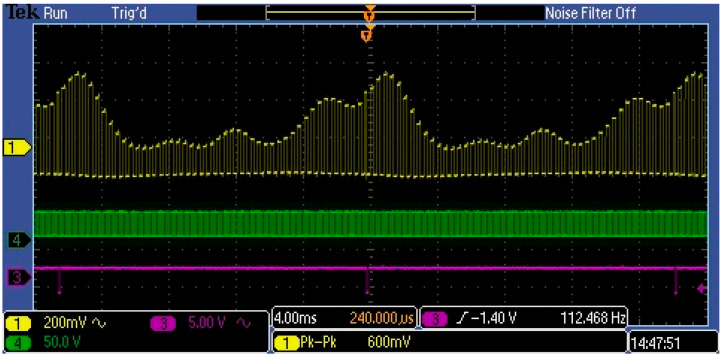
Captured variable ingredient in Video signal with four lighting LEDs.

**Figure 11 sensors-23-03285-f011:**
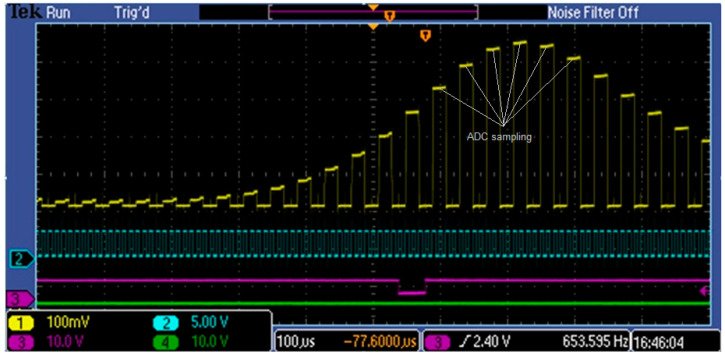
Video pulse sampling points.

**Figure 12 sensors-23-03285-f012:**
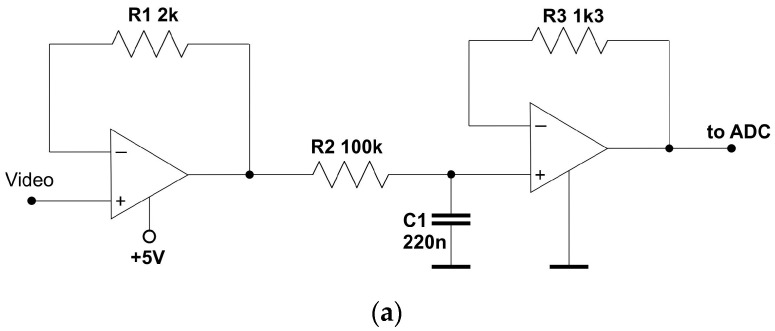
Video signal filters: (**a**) DC low-pass filter; (**b**) AC high-pass filter.

**Figure 13 sensors-23-03285-f013:**
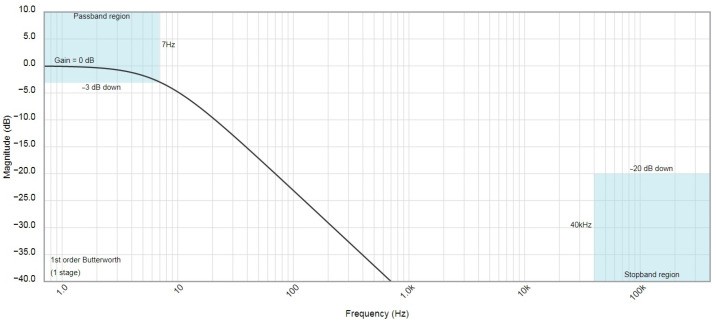
Amplitude frequency characteristics of the low-pass filter.

**Figure 14 sensors-23-03285-f014:**
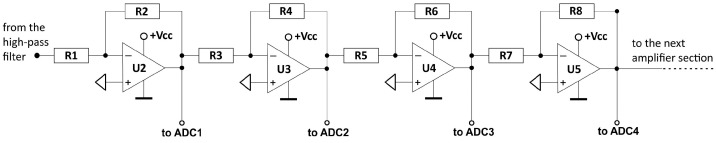
Diagram of the four-section ripple-curry amplifier.

**Figure 15 sensors-23-03285-f015:**
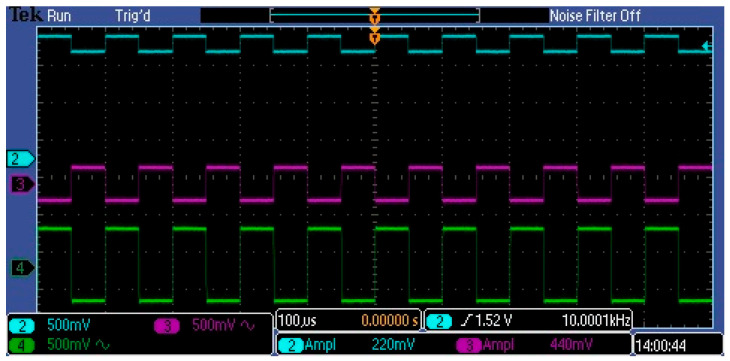
AC signal waveforms for the 10 kHz pulse frequency: blue waveform—input signal with high DC, purple—AC ingredient amplified twice, green—AC amplified four times.

**Figure 16 sensors-23-03285-f016:**
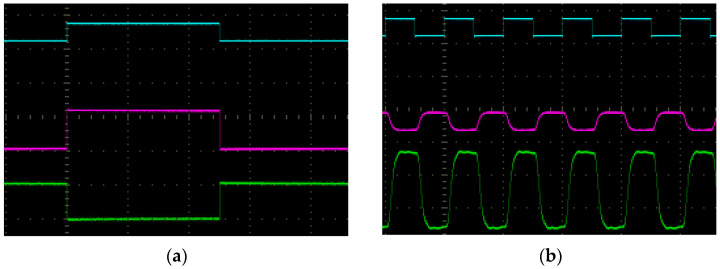
AC signal waveforms for the pulse frequency: (**a**) 1 kHz; (**b**) 100 kHz.

## Data Availability

The data is not publicly available because of continuing research work on the larger system, and the problem described in the article is the selected issue of the system being developed.

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
