# Peer review of "Selective Light Measurement in the Control of Reference LED Sources"

_sensors, 2023, doi:10.3390/s23063285_

Round 1

Reviewer 1 Report

My general evaluation of the study titled "Selective Light Measurement in Control Systems" is as follows.

   It is a study in the field of “MEMS optical sensor in the applied research of LED lamps”. It is seen that the study was organized and written in accordance with its purpose. Making the following corrections in this study will make the article stronger.

1. The abstract should be revised. The technique-method can be stated briefly. The purpose is not clearly stated.

2. In general, the English language of this article should be corrected. Professional help is recommended.

3. I don't think the Introduction section is sufficient. The number of resources should be increased. It is possible to benefit from the studies carried out in recent years.

4. It is recommended to use Figure 13,15,16 by arranging them in a single figure. You don't need to give too many details.

5. In the same way, you can use Figure 1,2,3 in a single figure. I think it is unnecessary for you to use too many graphics. Please remove unnecessary graphics in general.

6. In the Conclusions section, the superiority-difference of the article from other existing studies should be clearly stated. Please improve this section.

Author Response

Dear Sir/Madam,

I, thank you very much for your constructive comments, which I have tried to meet in the revised version.

I am at your disposal in case you need additional correction.

I slightly changed the title, which now clarifies the subject of the article.

I expanded the introduction and completed the literature review, which more broadly introduces the background of the issue.

I combined several drawings into one, removed some less significant ones and added one drawing that I consider important to the issue and the state of knowledge and technology in the subject area.

I have not drawn a complete schematic of the entire circuit, as it would be hardly visible.

If there is a need, I can add an electronic block diagram with description.

My article focuses on the issue of stabilization of LED light sources.

Such sources can be standard in measurement technology, or their circuit can be a hardware simulator of some illuminating lamps, which is useful during their design.

Particularly close to my heart is the second theme, i.e. the use of the circuit as a tool to support the synthesis of specialized lighting lamps.

In particular, this applies to greenhouse lamps, which I deal with, review projects related to them, and where I see a lot of mistakes made at the design stage.

I attempted to build such a simulator, and in the process revealed the problem of designing a miniature circuit for the real-time control of luminous flux in the simulator. I analyzed many of the spectrum measurement circuits in use and my choice focused on Hamamatsu's MEMS C12666MA spectrometer.

It is a verified commercial chip, which avoids the difficulties of implementing practically unverified concepts.

Studying the literature related to this microspectrometer, I noticed knowledge gaps, ambiguities in its operation, incomplete technical documentation and the related distrust of metrologists in its use.

Moreover, I found an article in the IEEE Sensors Journal [29], which incompletely describes the using of the C12666MA sensor and some of the solutions adopted in it I consider incorrect.

A separate critical article can be written on this subject, in my article I only signal the problem. The reader, analyzing the said article, may draw wrong conclusions and build a circuit with which he will have trouble. Such a condition was an additional motivation for writing my article.

My article mainly focuses on the issue of selective light measurement in an LED illuminator built on an integrating sphere. Such online measurement is necessary to correct the operating conditions of the LED illuminator in order to keep the luminous flux stable.

The design of the illuminator is currently under development, and the system for selective light measurement is only a part of it, which was the basis for the article.

All the modules of the system under construction, with the exception of the integration sphere and the FPGA control module and the C12666MA chip, I designed and fabricated myself. I do not have extensive facilities or technological support, hence the execution of the construction was within my capabilities.

I am open to additional suggestions and possible corrections.

My Best Regards,

Marian Gilewski

Reviewer 2 Report

Review of the manuscript entitled ‘Selective Light Measurement in Control Systems’

The motivation for the present work was the unsatisfactory state of the art in LED lamp applied research. Hence, the paper focuses only on the possibly compact solution for spectral light measurement in the system, without the need for external apparatus

1) ‘…they have lower measurement resolution, and measurements also had higher overall errors …’ How much?

2) Sometime the word MESM replaces MEMS.

3) The manuscript must be polished and English spellings corrected. As an example:

… described system is preferable to using a microcontroller…  

4) A complete electronic scheme of the system could help the reader to understand.

5) The final results with the LED spectrum is also interesting for the reader.

Author Response

(The authors gave the same response as above.)

Round 2

Reviewer 2 Report

The manuscript has been corrected and greatly improved. It can be accepted for publication.